# Development of Air Bearing Stage Using Flexure for Yaw Motion Compensation

Hak-Jun Lee [1] and Dahoon Ahn [2,*]

1. Smart Manufacturing System R&D Department, Korea Institute of Industrial Technology, Cheonan-si 31056, Korea; hak1414@kitech.re.kr
2. Department of Mechanical System Design Engineering, Seoul National University of Science and Technology, Seoul 01811, Korea
* Correspondence: dhahn@seoultech.ac.kr

**Abstract:** This paper presents an air bearing stage that uses flexure for yaw motion compensation. The proposed stage realizes motion in three degrees of freedom (DOF), which are the $X$, $Y$, and $\Theta_z$ directions. To work with $\Theta_z$ as the rotational motion of the stage, we applied a flexure consisting of four bar linkages. The stage from a previous study in which flexure is applied to compensate for yaw motion error has the limitation of increasing the structural stiffness of the stage due to the rotational stiffness. In this study, we propose a combination of a new stage structure and flexure to ensure the high structural stiffness of the stage and the very low rotational stiffness of the flexure at the same time. Modeling and design optimization were performed to apply adequate flexure to the proposed stage. Experiments were carried out to verify yaw motion error compensation and the performance of the stage. The proposed stage has a maximum yaw motion error of 0.86 arcsec during the scanning motion and 48 ms settling time, while the stepping motion is improved by 34.2% compared to the previous study.

**Keywords:** air bearing stage; error compensation; flexure; yaw motion

## 1. Introduction

Due to the automation of manufacturing processes, various types of equipment have been required and developed. They demand a long motion range and stable operation to handle a workpiece. Some manufacturing processes need to move a workpiece (e.g., a glass panel) on the XY plane, which is perpendicular to the gravity direction. Therefore, XY stages using linear motors and air bearings in particular are widely used to handle wafers for semiconductor manufacturing, glass panels for display manufacturing, and tools and probes for high-precision machining and measuring.

The XY stage usually employs several air bearings, utilizing their lack of friction for precision motion [1–4]. Linear motors are adopted as actuators for the same reason, usually one for the $X$-axis and two for the $Y$-axis, or vice versa [5–8]. The moving sliders, linear motors, and air bearings are all placed on a granite base as the reference guide plane of the air bearings. The XY stage is designed to precisely move the workpiece, but there is the inevitable issue of yaw motion error, which is the misalignment of the moving slider containing the workpiece with respect to the reference frame in the rotational direction ($\Theta_z$). Yaw motion error is a very important problem because it largely affects the final performance of the stage. Even a very small error should be eliminated, as it can lead to an exaggerated large disagreement on the outer edge of the workpiece. Nonetheless, efforts to remove the causes of yaw motion error at the level of fabrication or assembly are limited.

Yaw motion error can occur in response to the surface flatness of the guide plane of the granite base. It also occurs due to the assembly error of air bearings. Even though well-set-up air bearings try to make the sliders move in only the desired directions, parasitic motion is inevitable. Another major source of yaw motion error is the imperfect synchronization of





several linear motors driving the stage. The tolerance of the fabrication and assembly of the stage and the surface flatness of position feedback sensors also affect the yaw motion of the stage, and they cannot be decreased below a certain level.

One approach to addressing this problem is introducing another mechanism with more than one degree of freedom to the XY stage. Yaw motion for error compensation can be realized by simply adding a rotary stage to the existing stage. However, this makes the inertia of the moving body larger and the position of its mass center higher. A dual servo mechanism can be an option to resolve these issues, but additional actuators and guides lead to complex structure and control algorithms [9,10]. The cross-coupled control scheme by Giam et al. [11,12] can be applied to compensate for the yaw motion error of a gantry-type stage in real time. This method does not demand additional actuators or guides, and its performance has been proven for the XY stage using ball contact guides. It seems easy to apply this control scheme to an air bearing guided stage, but failure of the air bearing could be induced as a result. Therefore, a flexure is installed at the connection points between the sliders moving along the *Y*-axis and the crossbeam placed on the sliders, allowing relative rotational movement of the crossbeam to the sliders and the air bearings. J. Ma et al. [13,14] presented optimized designs of flexures, but contact-type guide mechanisms of the gantry axes were used. In a previous study, one degree of freedom (DOF) flexure was developed to accomplish rotational motion about the *Z*-axis [15]. The low stiffness of the flexure was favorable for yaw motion, but high stiffness of the flexure was also required to achieve the high structural stiffness of the stage. Thus, the optimal design to obtain sufficient yaw motion range resulted in a very low first structural mode frequency. An optimal design was also tested in another way to obtain high structural mode frequencies, but the yaw motion range was insufficient, and linear motors of very high force were required. This was due to the limitation of the flexure to achieve different stiffness in different directions by elastic deformation of one mechanical structure.

In this study, a high-precision air bearing guided XY stage utilizing flexure was developed and evaluated for yaw motion error compensation. A flexure was adopted to enable the yaw motion of the crossbeam and protect the air bearings from abrasion as well, while the yaw motion was achieved by the cross-coupled control scheme. The gap change of the air bearing of the conventional stage was quantitatively evaluated, and the necessity of the flexure was confirmed. A new structure of the stage, including a change in the installation positions of the linear motors and the flexure, was developed; this led to the achievement of very low rotational stiffness of the proposed flexure and very high structural stiffness of the stage at the same time. Thus, yaw motion error compensation can be accomplished by linear motors of moderate force.

The rest of this paper is organized as follows: Section 2 introduces the structure of the stage for yaw motion error compensation, and Section 3 details the design results of the developed flexure. Section 4 presents the experimental evaluation and discussion, and Section 5 presents the conclusion.

## 2. Design of Air Bearing Stage

This research targeted the maskless lithography process of a flat panel display (FPD) handling an 8th-generation glass panel with a size of 2200 mm × 2500 mm. In this paper, the designed stage is a pilot model for verification of yaw motion error compensation, and the motion range is 200 mm × 200 mm. It requires moving a glass panel on the XY plane, which is perpendicular to gravity. The target specifications are listed in Table 1. Since the design procedure and manufacturing technology of typical air bearing stages are very well known [16], this paper mainly describes the difference of the designed stage from conventional stages, focusing on the mechanism of yaw motion error compensation.

**Table 1.** Target specifications of the proposed air bearing stage.

|  | Unit | Specification |
| --- | --- | --- |
| Motion range | mm | 200 |
| In-position stability | μm | ±0.05 |
| Yaw motion error | arcsec | ±1.0 |
| Maximum speed | mm/s | 100 |
| Maximum acceleration | m/s$^2$ | 2 |
| Settling time 2 μm move, 1% (20 nm) | msec | 500 |

## 2.1. Stacked Gantry Structure

XY motion stages with air bearings have various structures, such as stack type [17], stack and open type [18,19], window type [20], and H-type [8,21,22]. The H-type structure is widely used due to its simple structure and the low-positioned center of mass of the moving components. The H-type structure has a common guide surface of air bearings for motion along the *X*-axis and *Y*-axis; Figure 1a,b show the schematic view of the H-type structure. Thus, the surface area of the guide should be four times larger than that of the workpiece; i.e., the surface area is 4400 mm × 5000 mm in the case of an 8th-generation FPD. However, it is impossible to fabricate such a large granite guide with fine surface finishing. An assembly of several guides might be an option, but the discrete borders of those surfaces are not suitable for air bearings. Therefore, the proposed XY stage has a stack-type structure employing three long granite blocks. Figure 1c,d shows the schematic view of the proposed XY stage. One of them is a crossbeam placed along the *X*-axis, and it is the guide surface of the X-slider that moves along the *X*-axis. The others are placed in parallel along the *Y*-axis, and the guide surfaces of the crossbeam and Y-sliders move along the *Y*-axis. Therefore, the proposed stage can be said to be designed as a stacked gantry structure.

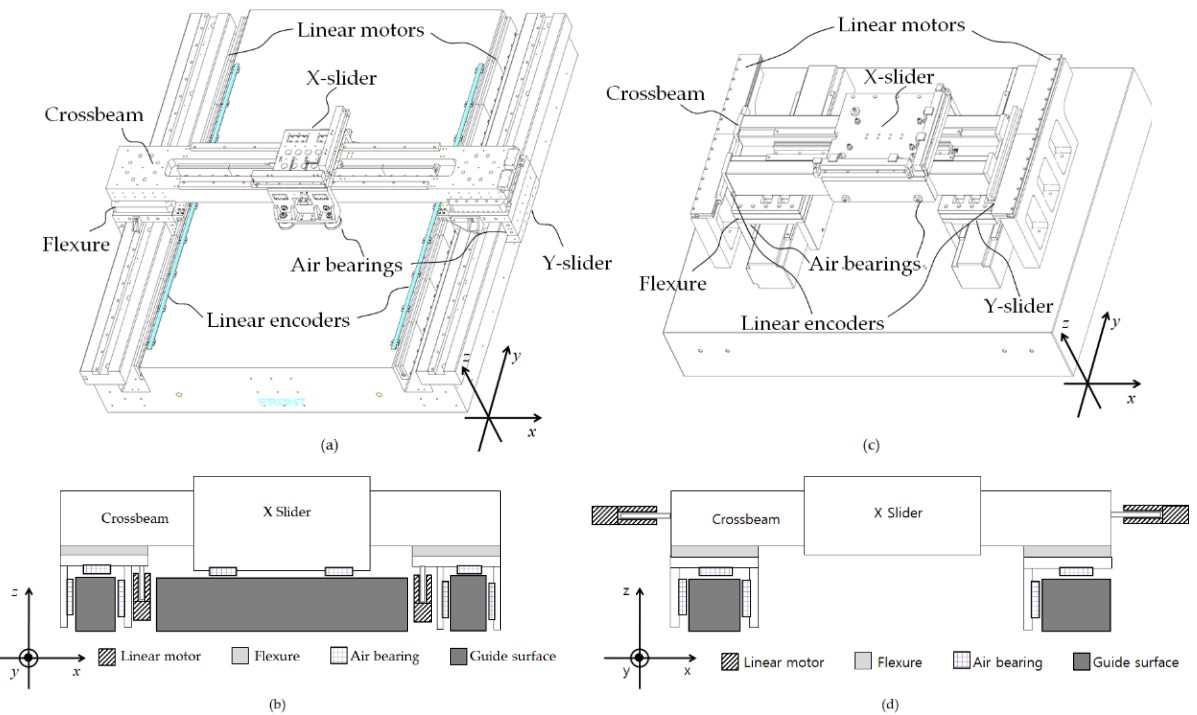

**Figure 1.** Schematic diagram of stages for yaw motion error compensation: (**a**) 3D drawing of the conventional H-type stage of the previous study [15], (**b**) schematic front view of the conventional H-type stage, (**c**) 3D drawing of the newly proposed stage of stack and gantry structure, and (**d**) schematic front view of the newly proposed stage.

Linear encoders (MS80, RSF Elektronik, Frankfurt, Germany) are used to provide feedback on the positions of the X-slider and Y-sliders, as shown in Figure 1c. Three linear encoders are precise enough to satisfy the specifications of Table 1. Two scales and laser heads measure the displacement of the Y-sliders and the yaw angle of the crossbeam, and the other scale and laser head measure the displacement of the X-slider. The position feedback provided by the linear encoders offers the advantage of servoing the stage in a real-time operation, leading to reduced yaw and straightness error.

### 2.2. Air Bearings and Flexure

The moving components of the stage have a number of air bearing pads to guide themselves in the desired motion directions, as shown in Figure 1a,c. Thus, the arrangement of the pads should provide the structure with high stiffness in all directions except the direction of motion. The developed stage utilizes 18 air bearing pads (EZ-0053, Eitzenberger Luftlagertechnik GmbH, Munich, Germany) with preload generated by permanent magnets and steel rods. The air bearings should maintain an air gap of around 5 μm from the guide surface for normal operation. For the guide surfaces placed along the *Y*-axis, two granite blocks with twice the length of the working range are used. Since it is not possible to achieve parallelism within 5 μm of the two granite blocks, one is designated the master guide surface, and the moving components follow it using magnetic preload. The other one just provides a guide surface as a slave. This technique is the master–slave guide, which is often used to ensure the precision of air bearing stages.

In the case of a typical air bearing stage without flexure, the air gap of the air bearing becomes large, inducing functional failure of the air bearings when the crossbeam rotates for yaw motion error compensation. Equation (1) and Figure 2a show the variation in the air gap when the crossbeam rotates to compensate for yaw motion error. If the slider rotates 40 arcsec, the air gap change of the air bearing is about 9 μm. It is very large compared to the operating range of the air bearing pads. The gap changes can cause the bearing to move away from the guide surface or make contact with the guide surface, which can disrupt dynamic performance or cause damage to the air bearing. Thus, we propose using a flexure to permit a very small rotation between the crossbeam and the two Y-sliders. The change in the air gap with the flexure is shown in Figure 2b and Equation (2) when the flexure has an ideal rotational stiffness of zero.

$$\Delta = L \cos \Phi - L \cos(\Phi + \Theta) \approx 9 \ \mu m \tag{1}$$

$$\Gamma = L_f - L_f \cos \Theta \approx 0.25 \ nm \tag{2}$$

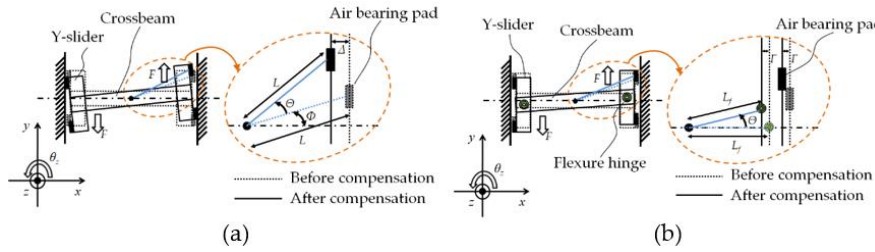

**Figure 2.** Change in gap between the air bearing pad and guide surface when the yaw motion error is compensated; (**a**) air bearing stage without a flexure; (**b**) air bearing stage with flexures.

L and Φ represent the position of the air bearing pad relative to the center of the crossbeam. L is 470 mm, and Φ is 0.325 rad. $L_f$ denotes the distance from the center of the crossbeam to the center of the flexure and is 383.75 mm. Θ is the yaw motion of the crossbeam. As seen from (1) and (2), the flexure eliminates the risk of damage to the pads and the guide surface or functional failure of the air bearings when compensating for the yaw motion error.

*2.3. Linear Motor and Flexure*

The proposed stage has two linear motors (SGSGW-60A365C, Yaskawa, Kitakyushu, Japan) to drive the crossbeam along the *Y*-axis and one linear motor to drive the *X*-slider along the *X*-axis (SGLGW-40A253C, Yaskawa, Japan). The installation positions of the linear motors are shown in Figure 1d.

In the case of the conventional stage, as shown in Figure 1b, the stator of the linear motor fixed to the base drives the mover fixed to the Y-slider. This is a typical construction that is widely used for XY stages, where the crossbeam and the Y-sliders can be considered to be one rigid body. They are connected by fasteners. The air bearing stage of the previous study adopted the typical installation position of the linear motors. However, the air bearing stage compensating for yaw motion error had flexures between the crossbeam and the Y-slider to prevent failure of the air bearings. When the linear motors drove the Y-sliders with flexures, the crossbeam was pulled with the Y-sliders through the flexures. Therefore, the flexure should enable rigidity of the stage structure to ensure good dynamic performance, but this is not easy because the flexure should also be flexible enough to rotate for yaw motion error compensation. In the case of the proposed stage, the movers of the linear motors are placed on the crossbeam. This can lead to high structural stiffness of the stage and a rotationally flexible flexure at the same time. The linear motors drive the crossbeam directly, and the Y-sliders are pulled with the crossbeam. Since the yaw motion of the crossbeam is directly accomplished by the linear motors, low structural stiffness of yaw motion is not a problem. The yaw motion mode of the proposed stage was controllable, but that of the conventional stage was not. Thus, the limit to lower the rotational stiffness of the flexure was removed by the change in the stage structure. The comparison of the conventional and newly proposed stages is shown in Figure 3.

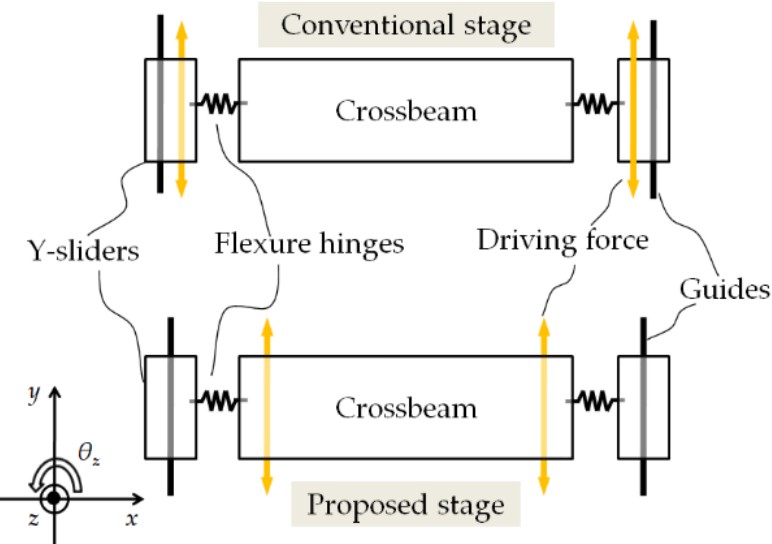

**Figure 3.** Difference in structure between the conventional stage and the newly proposed stage.

## 3. Design of Flexure

*3.1. Conceptual Design*

As mentioned in the previous section, the flexure should have low stiffness in the rotational z-direction and high stiffness in the other five directions. The lower the rotational stiffness, the less force required by the linear motor; the higher the stiffness in the other directions, the higher the structural rigidity and mode frequency of the stage achieved. Less force of the linear motor leads to less power consumption in operation, especially for holding the stage in position and maintaining the yaw angle. To meet the design requirement, we applied a four-bar linkage structure as well as circular notch hinges to the flexure. Figure 4a shows the skeleton diagram and 3D conceptual design of the

proposed flexure for one rotational motion. The four-bar linkage structure has low stiffness in one direction and high stiffness in the other directions. As shown in Figure 4a, the intermediate bodies, which are hatched squares in the skeleton diagram, were connected to the crossbeam, and the base of the flexure was fixed to the Y-slider. The linkage structures are arranged in a circle around the center to have low stiffness in the rotational z-direction and high stiffness in the other direction. As seen from the conceptual design of the flexure in Figure 4a, different stiffness in different directions is needed for one elastic component; hence, an optimal design process is needed to determine geometrical parameters and achieve adequate stiffness of the flexure.

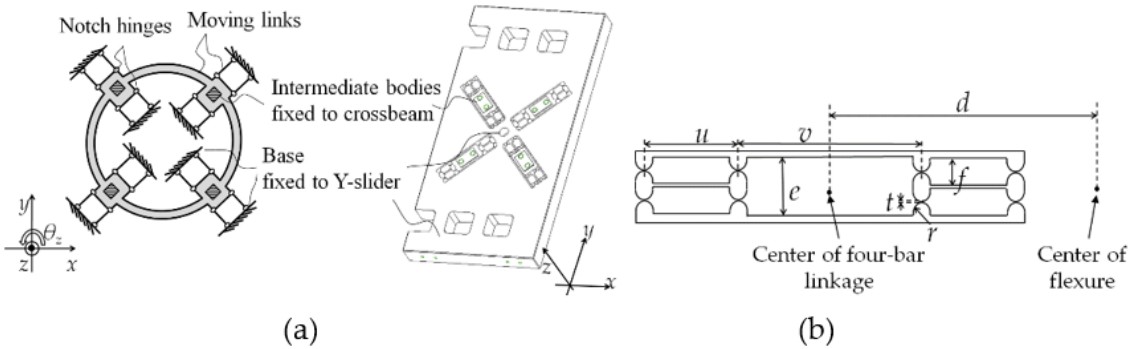

**Figure 4.** (**a**) Skeleton diagram (left) and 3D conceptual design (right) of the flexure for one rotational DOF motion; (**b**) design parameters of the proposed flexure.

### 3.2. Mathematical Modeling

For the optimization of the proposed flexure, a mathematical model explaining the relationship between the stiffness and the geometric dimensions of the flexure was established. The mathematical model can be obtained from the motion equation of the flexure, regarding the 17 links as rigid bodies and the 32 notch hinges as massless spring components. Therefore, a $6 \times 6$ compliance matrix of a notch hinge is required, and the model by Koseki et al. [23] was used in this study, as shown in (3) and Table 2, where E and G are the elastic and shear moduli of the flexure material.

$$
\begin{bmatrix} \delta_x \\ \delta_y \\ \delta_z \\ \theta_x \\ \theta_y \\ \theta_z \end{bmatrix} = \begin{bmatrix} c_1 & 0 & 0 & 0 & c_3 & 0 \\ 0 & c_2 & 0 & -c_4 & 0 & 0 \\ 0 & 0 & c_5 & 0 & 0 & 0 \\ 0 & -c_4 & 0 & c_6 & 0 & 0 \\ c_3 & 0 & 0 & 0 & c_7 & 0 \\ 0 & 0 & 0 & 0 & 0 & c_8 \end{bmatrix} \begin{bmatrix} f_x \\ f_y \\ f_z \\ M_x \\ M_y \\ M_z \end{bmatrix}. \tag{3}
$$

**Table 2.** Compliance of notch hinge.

| Notation | Compliance |
|:---:|:---:|
| $c_1$ | $\left(9\pi r^{\frac{5}{2}}\right)/\left(2Ebt^{\frac{5}{2}}\right) + \left(3\pi r^{\frac{3}{2}}\right)/\left(2Ebt^{\frac{3}{2}}\right)$ |
| $c_2$ | $(12\pi r^2)/(Eb^3)\left\{(r/t)^{\frac{1}{2}} - 1/4\right\}$ |
| $c_3$ | $\left(9\pi r^{\frac{3}{2}}\right)/\left(2Ebt^{\frac{5}{2}}\right)$ |
| $c_4$ | $(12r)/(Eb^3)\left\{\pi(r/t)^{\frac{1}{2}} - (2+\pi)/2\right\}$ |
| $c_5$ | $1/(Eb)\left\{\pi(r/t)^{\frac{1}{2}} - \pi/2\right\}$ |
| $c_6$ | $(12/Eb^3)\left\{\pi(r/t)^{\frac{1}{2}} - (2+\pi)/2\right\}$ |
| $c_7$ | $\left(9\pi r^{\frac{1}{2}}\right)/\left(2Ebt^{\frac{5}{2}}\right)$ |
| $c_8$ | $\left(9\pi r^{\frac{1}{2}}\right)/\left(4Gbt^{\frac{5}{2}}\right)$ |

In order to obtain the motion equation in (4), Lagrange's equation [24–26] is used, and the compliance model of (3) is used to calculate potential energy stored in each notch hinge.

$$\mathbf{M}\ddot{x} + \mathbf{K}x = \mathbf{F} \tag{4}$$

$$\mathbf{M} = \begin{bmatrix} M^1 & & 0 \\ & \ddots & \\ 0 & & M^{17} \end{bmatrix}_{102 \times 102} \tag{5}$$

$$M^i = \begin{bmatrix} m^i & m^i & m^i & I_x^i & I_y^i & I_z^i \end{bmatrix} \tag{6}$$

$$\mathbf{K} = \begin{bmatrix} K_{11} & \cdots & K_{1,17} \\ \vdots & \ddots & \vdots \\ K_{17,1} & \cdots & K_{17,17} \end{bmatrix}_{102 \times 102} \tag{7}$$

$$K_{ij} = \begin{cases} \sum_{k=1}^{32} T_k^{iT} k_k T_k^i, & for \ i = j \\ -T_k^{iT} k_k T_k^j, & for \ i \neq j \end{cases} \tag{8}$$

$$\mathbf{F} = \begin{bmatrix} f^1 & , \cdots, & f^i & , \cdots, & f^{17} \end{bmatrix}_{102 \times 1}^T \tag{9}$$

$$\mathbf{x} = \begin{bmatrix} q^1 & , \cdots, & q^i & , \cdots, & q^{17} \end{bmatrix}_{102 \times 1}^T \tag{10}$$

M and K are the mass matrix and stiffness matrix, respectively, of the flexure, and mi is the mass of the *i*th body. $I_x$, $I_y$, and $I_z$ are the moments of inertia with respect to local coordinate systems, as shown in (6). Since the number of links of one flexure is 17, M and K are 102 × 102 matrices, as shown in (5) and (7). Each element of (8) is the stiffness of each notch hinge in global coordinates. $T_k^i$ is the transformation matrix to change the coordinates from local to global. F is the force vector indicating all force components exerted on each link, as shown in (9). The displacement vector *x* is defined by (10), and qi is the displacement vector of the center point of the *i*th link, which can be expressed by (11).

$$q_i = \begin{bmatrix} x_i & y_i & z_i & \theta_{xi} & \theta_{yi} & \theta_{zi} \end{bmatrix}^T \tag{11}$$

If the link directly connected to the crossbeam is numbered "1", the directional stiffness of the flexure can be obtained from (12) by making all elements of the force vector zero except the first six elements.

$$\mathbf{x} = \mathbf{K}^{-1}\mathbf{F} \tag{12}$$

For example, if F is set to $[1, 0, 0, 0, 0, 0, \cdots \cdots, 0]^T$, the first element of x is the displacement of the link directly connected to the crossbeam along the *X*-axis when the unit force is applied along the *X*-axis. Then, it is the stiffness of the flexure for the direction along the *X*-axis or $K_x$. The stiffness in other directions, $K_y$, $K_z$, $K_{\theta x}$, $K_{\theta y}$, and $K_{\theta z}$, can be obtained in the same manner.

### 3.3. Optimization

Using the developed mathematical model, we optimized the flexure. The objective function was set as (13) in order to obtain low stiffness in the rotational z-direction and high stiffness in the other directions. w1 to w6 represent weighting factors to equalize the effects of different values of stiffness in each direction.

$$f_{obj} = min(w_1/K_x + w_2/K_y + w_3/K_z + w_4/K_{\theta_x} + w_5/K_{\theta_y} + w_6 K_{\theta_z}) \tag{13}$$

Designing adequate flexure for the proposed stage requires constructing design constraints as follows. The first constraint of the minimum rotational motion range was set to 80 arcsec. It was established by multiplying the safety factor of 2 by the alignment tolerance of the linear encoders, which is 40 arcsec. The change in deflection of the crossbeam in the

gravitational direction induced by the change in the workpiece should be smaller than 3 μm due to the operating range of the inspection probe. Third, the stiffness in the *X*-direction of the flexure should be large enough for the X-slider to settle quickly. When the X-slider is actuated by the maximum acceleration of 2 m/s$^2$, a reaction force of 45 N is exerted on each flexure. The stiffness of the flexure in the x-direction is constrained to 900 N/μm, which corresponds to deformation of 100 nm. Finally, the size of the flexure should be smaller than $200 \times 200 \times 28$ mm$^3$, and there should be no interference between the design parameters. The size is determined by the area of the Y-slider.

The design parameters of the proposed flexure are shown in Figure 4b, presenting one of four bar linkages; u and f are the length and width, respectively, of a link; v and e are the length and width, respectively, of the link connected to the crossbeam; d is the distance from the center of the flexure to the center of the four-bar linkage; and r and t are radius and thickness, respectively, of the notch hinge. All variables shown in Figure 4b were used in the design process and are independent of each other. The shape of the flexure can be determined only when all of the variables are set, and they are also involved in the behavior of the flexure. To determine the optimal design parameters, the sequential quadratic programming (SQP) method of MATLAB was used. We confirmed the convergence of the objective function. Table 3 presents the final design results of the proposed flexure. The values were determined considering the manufacturing tolerance. The optimal design result has lower rotational z-directional stiffness and higher stiffness in the other directions, just as we intended.

**Table 3.** Final design results.

| Parameters | Design Results (mm) |
|:---:|:---:|
| u | 19.0 |
| v | 48.8 |
| d | 63.4 |
| e | 23.0 |
| f | 10.5 |
| r | 3.2 |
| t | 0.6 |

To verify the optimal design results, we simulated the static stiffness of the proposed flexure using the finite element method (FEM) tool (Pro/Engineer, PTC, Boston, MA, USA). The 6-axis stiffness of the proposed flexure from the simulation was compared to the design results based on the analytical model. As shown in Table 4, the analytical model was found to be suitable to predict the actual behavior of the flexure.

**Table 4.** Verification of stiffness of flexure by FEM analysis.

| Stiffness | Modeling | FEM | Error (%) |
|:---:|:---:|:---:|:---:|
| $X$ (MN/m) | 976 | 996 | 2.01 |
| $Y$ (MN/m) | 976 | 949 | 2.85 |
| $Z$ (MN/m) | 492 | 523 | 5.93 |
| $\Theta_x$ (kNm/rad) | 1360 | 1453 | 6.4 |
| $\Theta_y$ (kNm/rad) | 1360 | 1406 | 3.27 |
| $\Theta_z$ (kNm/rad) | 86.3 | 83.1 | 3.85 |

## 4. Implementation of Stage

### 4.1. Manufacturing

Using the optimization results, the flexure was fabricated from AL 7075 material by wire-cut electrical discharge machining. Figure 5a shows the fabricated flexure. Figure 5b presents the whole air bearing stage where the fabricated flexure is installed. The developed stage measures 1500 mm × 1860 mm. Four passive isolators were installed to attenuate the ground vibration.

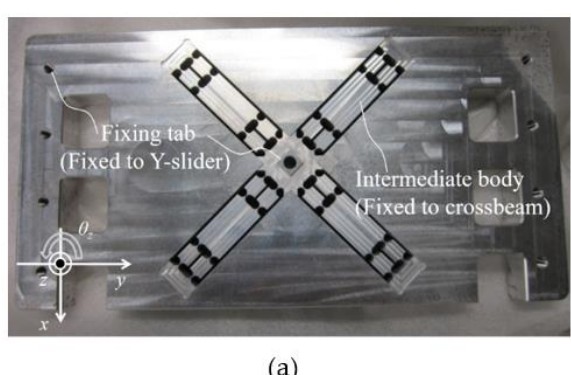 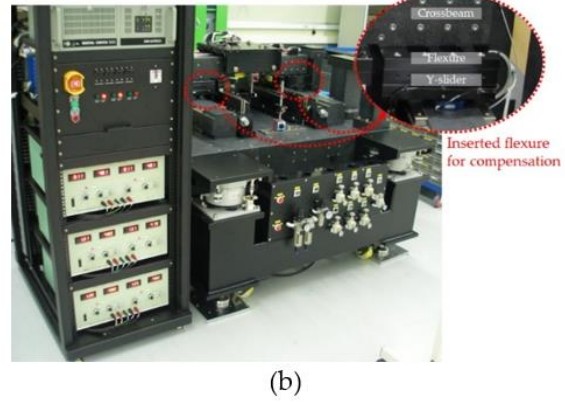

(a) (b)

**Figure 5.** (**a**) A manufactured flexure to be inserted between the crossbeam and the Y-slider; (**b**) air bearing stage where the proposed flexures are installed.

*4.2. Experimental Setup*

The developed air bearing stage was controlled using a real-time controller (DS 1005, dSPACE, Germany) and 3 current amplifiers (TA320, Trust Automation, San Luis Obispo, CA, USA). The current amplifiers apply current to the linear motors, which are integrated into the air bearing stage. The control strategy is described in Figure 6a. The cross-coupled control algorithm [12] was used, receiving position feedback from the 3 linear encoders representing the position of each linear motor. The signals from the linear encoders, $m_1$ to $m_3$, are transformed into the position feedback signals $x^m$, $y^m$, and $\theta_z{}^m$ using the sensor transformation matrix, as shown in Equation (14). The sensor transformation matrix is determined by the arrangement of the linear encoders. D is the distance between *Y1* and *Y2* encoders, as shown in Figure 6b. The position feedback signals represent the motion of the workpiece carrying the X-slider in the Cartesian coordinate system, which corresponds to the control axes. The proportional-integral-differential (PID) algorithm was used for each control axis, with the input signals to the linear motors determined from the control signals using simple kinematics, as shown in Equation (15). It was reported by Giam et al. [12] and Tan et al. [19] that the cross-coupled control scheme augmented by sliding mode control or a disturbance observer showed enhanced performance compared to conventional cross-coupled control adopting a PID algorithm. In addition, PID control schemes combined with a fuzzy controller [27] or an adaptive jerk controller [28] have been studied. However, such advanced algorithms are not easy to implement in the commercial low-level controllers generally used for wafer or display manufacturing devices. Since the air bearing stage in this study targeted the lithography process of flat panel displays, the developed system was evaluated by the cross-coupled control scheme with a simple PID algorithm. Equations (14) and (15) represent the cross-coupled control scheme. The feedback of two motors, $m_2$ and $m_3$, are used to create two control axes, $Y$ and $\Theta_z$. The corresponding feedback signals are $y^m$ and $\theta_z{}^m$, which are the sum and the difference after being divided by D of two feedback signals, respectively. The trajectory command, $y^{cmd}$ and $\theta_z{}^{cmd}$, are generated directly for the two control axes, $Y$ and $\Theta_z$. The PID control output is then split into two components, $u_{y1}$ and $u_{y2}$, one for each motor.

$$\begin{bmatrix} x^m \\ y^m \\ \theta_z^m \end{bmatrix} = \begin{bmatrix} 1 & 0 & 0 \\ 0 & 0.5 & 0.5 \\ 0 & 1/D & -1/D \end{bmatrix} \begin{bmatrix} m_1 \\ m_2 \\ m_3 \end{bmatrix} \tag{14}$$

$$\begin{bmatrix} u_x \\ u_{y_1} \\ u_{y_2} \end{bmatrix} = \begin{bmatrix} 1 & 0 & 0 \\ 0 & 1 & 1 \\ 0 & 1 & -1 \end{bmatrix} \begin{bmatrix} u_x \\ u_y \\ u_{\theta_z} \end{bmatrix} \tag{15}$$

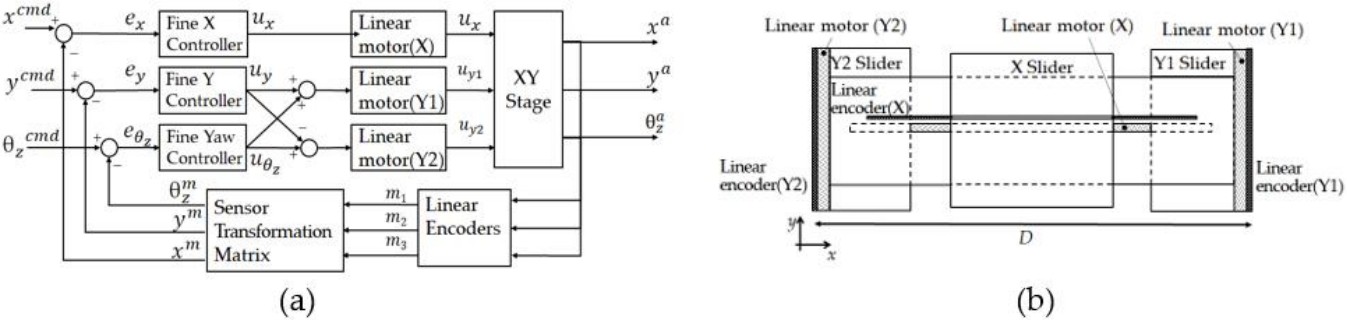

**Figure 6.** (**a**) Control algorithm for linear motion and yaw motion of the stage; (**b**) setup of linear motors and linear encoders.

### 4.3. Results

Based on the experimental setup, we evaluated the performance of the developed air bearing stage. First, the motion range, maximum speed, and in-position stability, which indicate the basic performance, were confirmed. Figure 7 shows the motion range and maximum speed of the air bearing stage. The blue line represents the displacement, and the red line represents the speed of the stage. The target values of 200 mm motion range and 100 mm/s speed were achieved for both axes. The position feedback signals received while the stage is held in position are shown in Figure 8. The in-position stability was evaluated with a standard deviation of 3, and the results are 0.0102 μm, 0.0088 μm, and 0.0079 arcsec for the $X$, $Y$, and $\Theta_z$ directions, respectively. Due to the stack structure of the stage, the motion of the X-slider is affected by the motion of the Y-sliders. Therefore, the in-position stability in the $X$-axis was slightly higher than that of the $Y$-axis. Based on the experimental results, the stage was found to have high precision in static conditions.

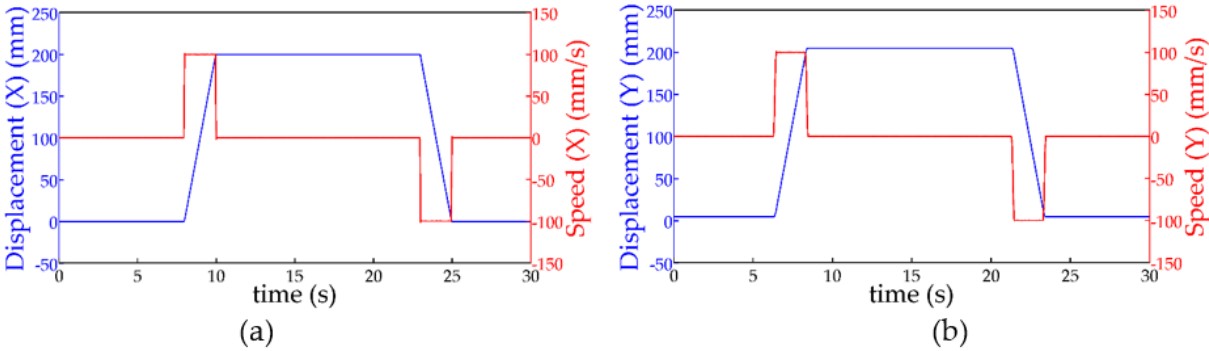

**Figure 7.** Experimental results of motion range and corresponding speed along (**a**) $X$-axis and (**b**) $Y$-axis.

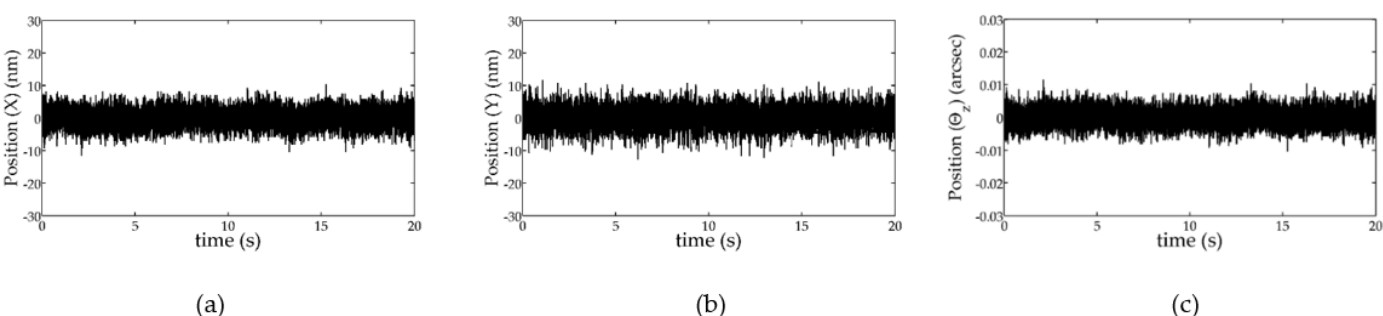

**Figure 8.** Experimental results of high-precision in-position stability: (**a**) $X$-axis, (**b**) $Y$-axis, and (**c**) $\Theta_z$-axis.

In this study, yaw motion error compensation was made possible by using the flexure and the two linear motors driving the crossbeam. For the verification of yaw motion error compensation in static and dynamic conditions, we performed a scanning motion experiment to check the currents of the *Y*-axis motors. Figure 9a shows the displacement and error of the scanning motion along the *Y*-axis. The stage moved forward at a constant speed, remained in position, and moved backward at a constant speed. Figure 9a,b show that the motion errors are small except for the moment when a jerk occurs. This means that the 3-DOF motion of the stage is well controlled while the stage is scanning and holding a position, except for moments of acceleration and deceleration. Checking the current inputs to the two linear motors (*Y1*, *Y2*) reveals that they are not equal and that they are not zero during motion at a constant speed. Since the air bearing guides are frictionless, the force required to keep the stage at constant speed is ideally zero. However, the current input of one linear motor is positive, and the other is negative. This means that the linear motors are trying to compensate for the yaw motion error during scanning. The current inputs are also not zero when the stage is holding a position. The linear motors apply forces corresponding to the restoring force of the flexures to keep the stage rotated and the flexure deformed. Figure 10 shows the result of yaw motion error measured in a global sense using a laser calibrator (ML10, Renishaw, London, UK) while the stage moves in the *X*- and *Y*-axis two times, forward and backward. The maximum yaw motion error during the motion was 0.86 arcsec. The target specification was confirmed to have been satisfied.

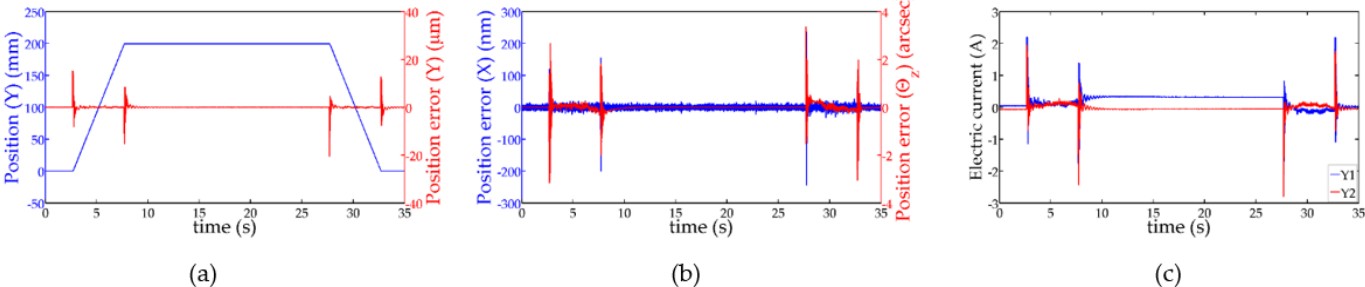

**Figure 9.** Experimental results of yaw motion error compensation during the scanning motion along the *Y*-axis, (**a**) position and error of the stage along the *Y*-axis, (**b**) position error of the stage along the *X*-axis and Θ$_z$-axis, and (**c**) electric current provided to the linear motors.

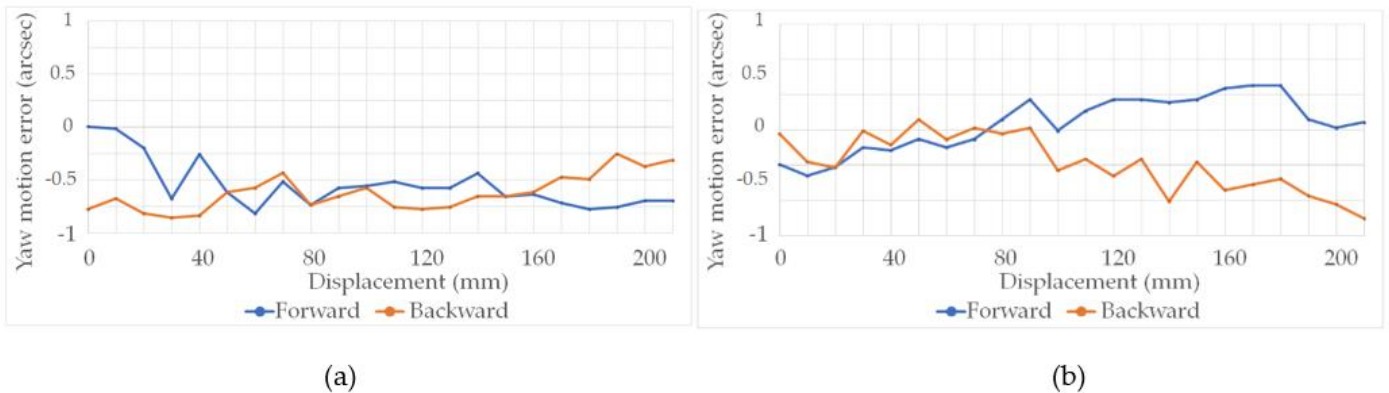

**Figure 10.** Measurement of yaw motion error: (**a**) *X*-axis motion and (**b**) *Y*-axis motion.

Finally, the dynamic performance of the stage was evaluated. Figure 11 shows the displacement and error of the stage when the motion command is a 2 μm step function. The blue line represents the position of the stage, and the red line denotes the position errors. The settling time was measured as the time taken for the position error to fall within 1% of the step size in order to check how quickly the point-to-point motion is accomplished. The measured settling time was 24 ms and 48 ms for the *X*-axis and *Y*-axis, respectively. The fast response is generally related to the overall rigidity of a system. A soft component

such as a flexure can lower the structural stiffness of a system. Nonetheless, the flexure of this study was optimally designed to have high stiffness in all directions other than the yaw motion direction. Moreover, the experimental results show that the yaw motion directly controlled by the linear motors is stiff enough to sufficiently satisfy the dynamic performance. A comparison to the data from the conventional air bearing stage from the previous study [15] is shown in Table 5, confirming the enhanced performance. The settling time and speed ripple were enhanced by approximately 34.2% and 57.6%, respectively.

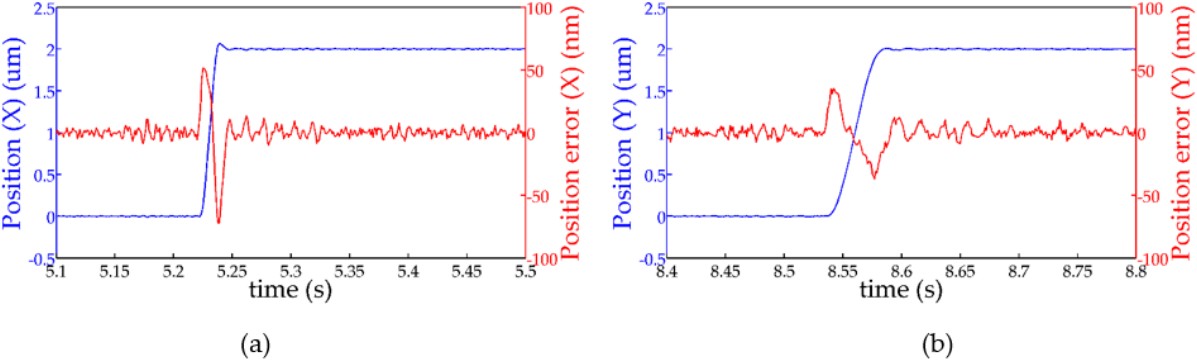

(a)                                                                              (b)

**Figure 11.** Experimental results of fast settling time: (**a**) *X*-axis, (**b**) *Y*-axis.

**Table 5.** Performance comparison to the conventional stage.

|  | Newly Developed Stage | Conventional Stage [20] |
|---|---|---|
| Settling time (2 μm step, 1%) | 48 ms | 73 ms |
| Speed ripple (100 mm/s) | 0.127 mm/s | 0.3 mm/s |

## 5. Conclusions

In this study, a new flexure and new structure were proposed for the yaw motion error compensation of an air bearing stage performing linear motion on the XY plane. The linear motors of the new structure directly control the yaw motion by applying force to the crossbeam. Due to the new structure, the air bearing stage has low stiffness to allow yaw motion for error compensation while having high structural stiffness in the other direction. Having low stiffness in the yaw motion direction enables a reduction in both the force of the linear motor and power consumption during linear motion or when stopping. Notch hinges were used to develop flexures with different stiffness for each direction, and design optimization of the flexure was performed. After verifying the accuracy of the design results through FEM analysis, an air bearing stage with flexure was fabricated. The result of the experimental evaluation of the manufactured air bearing stage showed a high precision of 10 nm and fast settling performance of 48 ms. In addition, it was confirmed that a non-zero current was applied to the *Y1* and *Y2* axes to compensate for the yaw motion error in a 200 mm linear motion or in a stationary state. Whereas previous studies reported the optimization of a flexure hinge without ensuring dynamic yaw motion error compensation performance due to the limitation of the stage driving force transmission method, this study verified the validation of the XY precision stage system using a new flexure and stage structure. The yaw motion error was compensated while preventing damage to the air bearing. At the same time, it was confirmed that the dynamic performance was enhanced compared to the previous study.

**Author Contributions:** Conceptualization, D.A.; methodology, D.A.; software, D.A.; validation, D.A.; formal analysis, D.A.; investigation, D.A.; data curation, H.-J.L. and D.A.; writing—original draft preparation, H.-J.L.; writing—review and editing, D.A. and H.-J.L.; visualization, H.-J.L.; supervision, D.A.; project administration, H.-J.L.; funding acquisition, H.-J.L. All authors have read and agreed to the published version of the manuscript.

**Funding:** This work is partly supported by the Korea Agency for Infrastructure Technology Advancement (KAIA) grant funded by the Ministry of Land, Infrastructure and Transport (Grant 20CTAP-C157468-02) and This study has been conducted with the support of the Korea Institute of Industrial Technology as "Development of holonic manufacturing system for future industrial environment (kitech EO-22-0010)".

**Institutional Review Board Statement:** Not applicable.

**Informed Consent Statement:** Not applicable.

**Data Availability Statement:** Not applicable.

**Conflicts of Interest:** Authors declare no conflict of interest.

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
