# Peer review of "Development of Air Bearing Stage Using Flexure for Yaw Motion Compensation"

_actuators, doi:10.3390/act11040100_

Round 1

Reviewer 1 Report

In this manuscript, Lee and Ahn presented the development of an air bearing stage using flexures, their proposed stage has an excellent yaw motion error and demonstrated significant improvements compared to the previous studies.

The paper is overall well-written and the figures are very clearly presented. The reviewer can recommend for publication with some minor suggestions.

Figure 1. It appears that Fig. (c) and (d) are cited before (a) and (b). As they represent the conventional design, they should be moved before the new design and are listed as (a) and (b) instead.

Figure 4(a). Font size should be increased.

Figure 5(b). The text in the zoomed-in box is difficult to read, please increase the font size.

Page 9, line 316. '... as shown in (14)', what is (14)? Line 322 shares the same issue.

Table 5 compares the settling time and speed ripple of the new and conventional stages.

The reviewer recommends to compare more essential data if available to demonstrate the advantage of the new stage.

Reviewer 2 Report

  • Good presentation of the research area;
    - Good presentation of the theoretical aspects;
    - Adequacy of the abstract;
    - Good presentation of the results and dissemination;
    - Appropriate references;
  • The conclusions could be more detailed:

- a comparison of the values of the error compensation ​​for the initial and the new structure would be useful

- How was compensate the yaw motion error for preventing air bearing failure

Reviewer 3 Report

This work designed a novel stage structure of air bearing stage to reduce yaw motion error. The authors devised flexure structure for the purpose. The design was compared with FEA simulation, and verified by  experiments. The work property has satisfied target specifications. Through design-simulation-experiment, the authors presented the novel XY stage. I think the manuscript can be accepted if some modifications are made.

In table 1, how were these specification values defined?
Especially, in introduction, the authors should explain how much of a number is a large yaw motion error.
Quantitative comparison between this specification of yaw motion error and values reached by conventional machines will more clarify the importance of this study.

Some variables in sentences are mismatched to variables in Figures.
For example, xm, ym, and θzm (at line 315) should be corrected using subscript.
I also found some others.

Equations (14) and (15) are missing?

Reviewer 4 Report

The article is written clearly and presents an interesting scientific solution regarding the yaw motion compensation of an air bearing stage. With the help of the newly added compliant four-bar linkages, the authors have successfully achieved low stiffness of the air bearing stage that allow yaw motion for error compensation and at the same time provides high structural stiffness in the other direction. The experiments were performed correctly and were confirmed experimentally.

In my opinion, the article could be improved if in Figure 4 (b) can shown more clearly which units are moving and which units are relatively stationary.
